# Chromosome Ordinal Number-Related Genomic Stability Revealed Among *Oryza* and Other Poaceae Plants

**DOI:** 10.3390/ijms26104778

**Published:** 2025-05-16

**Authors:** Xiyin Wang, Quanlong Liu, Bowen Song, Jiangli Wang, Wei Wang, Huilong Qi, Huizhe Zhang, Yuelong Jia, Yingjie Li, Zongjin Li, Miaoyu Tian, Yixin Cao, Yongchao Jin

**Affiliations:** 1School of Life Science, North China University of Science and Technology, Tangshan 063210, China; lql040919@163.com (Q.L.); baxigui2025@163.com (B.S.); qihuilong_9900@163.com (H.Q.); 18330525383@163.com (H.Z.); 15369408525@163.com (Y.J.); 15939950581@163.com (M.T.); cyxxxx050315@139.com (Y.C.); 2College of Mathematics and Science, North China University of Science and Technology, Tangshan 063210, China; 15732092893@163.com (Y.L.); lizongjin@ncst.edu.cn (Z.L.); jinyongchao@ncst.edu.cn (Y.J.); 3School of Public Health and Protective Medicine, North China University of Science and Technology, Tangshan 063210, China; 18093830463@163.com; 4Jitang Institue, North China University of Science and Technology, Tangshan 063210, China; wangweiwang26@163.com

**Keywords:** *Oryza*, genome stability, genome duplication, tandem genes, chromosomes

## Abstract

Rice (*Oryza sativa*) is one of the key staple crops, providing food for nearly half of the world’s population. The past twenty years have seen significant advances in understanding *Oryza* species through genome sequencing efforts. However, the stability of *Oryza* genomes during their divergence has not been well characterized. Here, by performing gene collinearity and comparative genomics analysis, we selected ten *Oryza* species and three other Poaceae species to check their genome stability, with *Leersia perrieri* as the reference. Intra- and intergenomic analysis showed a ~30% difference in homologous block numbers and a 35.7% difference in collinear gene numbers per block, indicating that *Oryza* genomes have undergone extensive DNA permutations. Notably, we found that *Oryza* chromosomes with smaller ordinal numbers have often preserved larger percentages of genes, while those with bigger numbers have undergone more gene losses. This unique observation may be explained by elevated gene losses incurred by illegitimate or homoeologous recombination between homoeologous chromosomes produced by the grass-common tetraploidization (GCT) ~100 million years ago (Mya), e.g., Chro. 11 and 12. However, the lowered gene loss rates in Chro. 1–3 could be explained by earlier restriction of illegitimate recombination after the GCT due to there often being (larger) neo-chromosomes produced by the fusion of ancestral chromosomes. The enriched NBS-LRR (nucleotide-binding site and leucine-rich repeat) genes in chromosomes 11 and 12 are another explanation for the above observation. Further evidence was obtained from other Poaceae plants. Moreover, we revealed around twice as many differences in tandem genes and their densities among *Oryza* plants, further showing their divergent levels of genome stability. The present efforts may contribute to the understanding of the stability of the *Oryza* genome and its formation, evolution, and functional innovation.

## 1. Introduction

Rice (*O. sativa*) is one of the key staple crops, providing food for nearly half of the world’s population. Due to their economic value, a large number of modern *Oryza* species have had their full genomes sequenced and resequenced. The International *Oryza* Genome Sequencing Project was launched in 1998, and the draft whole genomes of *O. sativa* ssp. Jing (Japonica) and *O. sativa* ssp. Xian (Indica) were completed in 2002. In order to better understand *O. sativa*, in 2005, the University of Arizona announced the start of the *Oryza* Map Alignment Project, and they eventually completed the genome sequencing of 11 wild *Oryza* species. In 2009, the International *Oryza* Map Alignment Project, based on the *Oryza* Genome Project, was launched to determine the genome sequences of eight AA genome types and nine non-AA genome types of *Oryza* species [1].

These *Oryza* genome sequences provide precious opportunities to explore the stability and plasticity of their genomes. Their genome stability and plasticity are related to the production and loss of genes, especially duplicated genes produced during polyploidization and through other types of gene duplication. The *Oryza* genomes shared a grass-common tetraploidization event (namely the GCT) that occurred ~100 million years ago. Polyploidy is considered to be a key driving force of species diversity and evolution. During polyploidization, all chromosomes and genes residing on them are duplicated. Afterward, the duplicated genome is often subjected to genome instability, and thousands of duplicated genes, including one copy or both paralogous copies derived from an ancestral gene, might be lost over a short period. A statistical model estimated that more than 90% of duplicated genes in the rice genome have been lost since the GCT event. During genome turmoil, genomic DNA might be extensively reorganized and chromosomes might be reduced. Imbalanced losses of genes may result in varied gene contents in different offsprings, leading to fast divergence. Eventually, these large-scale genomic changes could establish a new and large plant group. Notably, the Poaceae family, the Cucurbitecear family, the Legumiceae order, the major clade of the eudicot plants, that of the monocot plants, all the angiosperms, and even all the seed plants are each related to ancestral polyploidization [2,3,4,5,6,7,8].

The effects of ancient polyploidization could last for tens of millions of years. In the extant rice genome, thousands of duplicated genes produced by the GCT are retained, and similar observations have been made for the other *Oryza* genomes. These duplicated genes act as important genetic materials to set up novel pathways or rewire old genetic pathways. For example, tens of millions of years after the tetraploidization, the establishment of the C4 regulatory pathway involved the GCT-produced duplicated genes. Notably, the GCT-duplicated genes residing on individual terminal regions of rice chromosomes 11 and 12 have undergone prominent gene losses and ongoing gene conversion. Gene conversion was also inferred between other homoeologous chromosomes and chromosomal regions. Two functional paralogous genes were revealed on the terminal regions of rice chromosomes 11 and 12 [9]. They show the instability of the extant *Oryza* genomes and that very ancient duplicated chromosomal regions are still sources of genetic novelty.

Tandem genes could also lead to the instability of a genome [10]. With the exception of polyploidy-derived duplicated genes, many duplicated genes are in tandem locations on chromosomes, forming clusters containing several to tens of copies. In rice, 13% of genes have tandem paralogs. Ectopic DNA recombination could occur between tandem genes, resulting in the expansion or shrinkage of gene clusters and contributing to genome instability. Many NBS (nucleotide-binding site) disease resistance genes form tandem clusters, and genome instability due to intergenic DNA recombination provides a genetic force that produces novel copies with various mutations, such as point mutations or InDels. Alternatively, copies of genes could be lost during intergenic recombination. It was reported that tens of disease-resistance genes were lost in one generation of maize [11]. These facts show the genome instability produced by tandem genes. By analyzing the distribution, number, and sequence changes of tandem repeats, it is possible to reveal the replication, rearrangement, and evolutionary processes within the genome, as well as genomic differences among different species [12].

Like a coin with two sides, genome instability is also directly related to genome plasticity. Though genomes of species could be unstable due to polyploidizaton or the interaction, expansion, and/or contraction of tandem genes, many species have survived these often large-scale changes [13]. However, the plasticity and stability of *Oryza* genomes in the process of their divergence still need to be explored. Here, we selected ten *Oryza* species and three other species with completed genome sequences, including cultivars and wild species, sharing the same numbers of chromosomes (2n = 24), to check and compare their genome stability and plasticity. *L. perrieri* was used as an outgroup to infer gene collinearity and identify gene losses. Firstly, a list of homologous information of *Oryza* associated with polyploidy and species differentiation was constructed through comparative genomic analysis, and based on this, a list of ancestral genes of *Oryza* was constructed. Then, based on the results of genomic collinearity analysis, comparative genomics and phylogenetic genomics were used to reconstruct the phylogenetic relationships among *Oryza* plants. Furthermore, by constructing the list of tandem repeat genes of *Oryza* and combining with the list of ancestral genes of *Oryza,* the patterns of gene retention and loss among *Oryza* plants were analyzed to reveal the genomic plasticity and stability of *Oryza* plants. This study can deepen the understanding of *Oryza* plant divergence, genomic plasticity, and stability.

## 2. Results

### 2.1. Genome Instability Revealed by Characterizing Losses of GCT-Produced Paralogous Genes

Inferring intragenomic chromosome homology revealed ~5922–8848 collinear genes (Table 1), mostly paralogs produced by the GCT ~100 million years ago. The most collinear genes were observed in *O. punctata*, followed by *O. rufipogon* (8387) and *O. barthii* (8254), in contrast to the fewest in *O. meridionalis* (6843), followed by *O. nivara* (7005), and *O. brachyantha* (7038). The revealed collinear genes were distributed in 292–410 homologous blocks between relevant chromosomes. Though *O. glaberrrima* had around the average number of collinear genes among the studied plants, these collinear genes were found to be distributed across 410 homologous blocks, with an average of 19.2 collinear genes per homologous block, showing prominent genomic fractionation after the GCT. The fewest collinear genes per homologous block (18.3 or 7005 per 372 block) were found in *O. nivara*, showing also a prominent genomic fractionation level. In contrast, *O. barthii* and *O. rufipogon* had the lowest levels of genomic fractionation regarding collinear gene numbers per homologous block (25.6 and 24.1, respectively), supporting their higher genome stability after the GCT.

### 2.2. Genome Instability Revealed by Referring to the Outgroup Plant, L. perrieri

To assess gene retention after the split from the referenced species, we characterized the gene collinearity of each *Oryza* plant in comparison to the reference species. While aligning to the referenced genome, the *Oryza* genomes had between 499 and 1295 intergenomic collinear blocks and 14,924 and 19,203 collinear genes (Table 2).

Among all the *Oryza* plants, *O. nivara* had the most intergenomic collinear blocks (1295 collinear blocks), about two times the number found in *O. sativa* (Jing rice) (645 collinear blocks). In contrast, the contained collinear genes in *O. nivara* (16,143) were fewer than that in Jing rice (17,928). More collinear blocks may suggest a higher level of DNA fractionation after the split from *L. perrieri*. Therefore, the genome of *O. nivara* is the most unstable one due to its highest number of intergenomic collinear blocks. The fewest intergenomic collinear genes (12.5) per block also support this inference. In contrast, *O. brachyantha* had the fewest intergenomic blocks (499) and the densest collinear genes per block (32.88), indicating a relatively stable genome (Table 2).

The instability of a genome might be related to the accumulation of transposable elements (TEs), and the resulting DNA inversions by TEs may break collinear blocks into pieces. Based on the number of blocks compared to *L. perrieri*, the most instable genomes are those of *O. nivara* and *O. meridionalis*, while the most stable ones are those of *O. brachyantha*, *O. glaberrima*, and *O. sativa*. This is because the higher number of blocks in a genome indicates more genomic DNA rearrangements, which break gene collinearity into smaller blocks. Accordingly, the average number of collinear genes per block can also indicate the stability of genomes, suggesting that the most stable ones are those of *O. brachyantha*, *O. sativa*, and *O. glaberrima*, and the most unstable ones are those of *O. nivara*, *O. meridionalis*, and *O. glumaepatula*. These two criteria lead to a similar conclusion.

Gene retention/loss analysis was performed by aligning *Oryza* genomes (chromosomes) onto the referenced genome using the above-inferred intergenomic gene collinearity. We visualized the intergenomic collinearity (Figure 1) to show the gene retention/loss of the compared orthologous chromosome. Among *Oryza* plants, the highest gene loss rate occurred in *O. meridionalis* (51.7%), followed by *O. glaberrima* (46.6%) and *O. nivara* (46.3%), while the lowest gene loss rate occurred in Xian rice (35.0%), followed by *O. rufipogon* (36.2%) and *O. punctata* (37.0%). These findings show the instability of the former genomes.

Chromosomes show varying gene loss rates. An interesting finding was that chromosomes with smaller chromosome numbers had smaller gene loss rates, that is, genes on them were more likely to be retained in extant chromosomes. Regarding the gene content in the referenced genome, ~35% of genes in Chromosomes 1, 2, and 3 from the *Oryza* plants were lost, and the lowest loss rates occurred in *O. rufipogon* and *O. punctata* (Figure 2). The highest gene loss rates occurred in Chromosomes 11 and 12, reaching averages of up to 55% and 51%, respectively. In *O. meridionalis*, the gene loss rates in Chromosomes 11 and 12 reached up to 61% and 63% (Figure 2). These findings show that chromosomes with higher numbers suffered more gene losses, showing their instable nature, while chromosomes with smaller numbers were affected by fewer gene losses, indicating that they are more stable. The patterns discovered here are consistent among different *Oryza* plants. Furthermore, to analyze the gene loss rates between the homoeologous chromosomes and chromosomal regions, we separated the copies with more genes lost and those with fewer lost into two subgroups, as previously performed in the analysis of other ancient polyploids [14]. The subgroups can be regarded as subgenomes: one has preserved fewer genes, with 40–80% duplicated genes lost (Figure 2b), and the other has preserved more genes, with 10–50% duplicated genes lost (Figure 2c). This reflects the allo-tetraploid nature of the grass-common ancestor. These findings of two subgenomes are similar to the two revealed subgenomes derived from the tetraploid ancestor of the present maize species [15,16]. Additionally, the findings here about two rice subgenomes with diverged rates of gene retention (and gene loss) imply that allotetraploid genomes, although likely diverged when originated, could be subjected to instability tens of millions of years later.

### 2.3. Different Gene Loss Rates During the Divergence of Oryza Genomes

The gene loss rate at the time of species divergence may reflect the rate of genomic changes during evolutionary divergence, shedding light on the species’ ability to adapt to environmental changes and the magnitude of evolutionary pressures. We used the synonymous nucleotide substitution rate (Ks) among ten *Oryza* species, three other species, and the outgroup *L. perrieri* to estimate gene loss during their evolutionary processes. Based on the Ks, it can be inferred that the divergence of *L. perrieri* from the *Oryza* genus occurred 34–35 Mya, and the divergence between *O. brachyantha* and the other *Oryza* plants occurred 29–30 Mya. The divergence of the rice (*O. sativa*) from *O. rufipogon* occurred 0.9–1.0 Mya (Figure 3c). These estimated divergence times are similar to those reported by [17]. By checking the gene collinearity to the referenced *L. perrieri* genome, each *Oryza* plant lost 10,407–14,205 genes. We revealed that the loss of genes did not occur in a uniformed manner over time. After the split of *Oryza* plants from the referenced *L. perrieri*, and before the split of *O. brachyantha* and the other *Oryza* plants, in an estimated ~5 million years, 7354 genes were lost, resulting in a gene loss rate of 1470.8 genes per million years (Figure 3a). A prominent gene loss pattern, in number and loss rate, occurred before the divergence of *O. rufipogon* and *O. sativa*. A total of 417 genes were lost in ~0.36 million years before their divergence, resulting in a gene loss rate of 1158 genes per million years. Notably, after the split of *O. sativa* and *O. rufipogon*, the former seems to have lost 2193 genes, while the latter has lost 1229 genes, totaling 3602 gene losses per million years. That is, *O. sativa* ssp. Jing shows a 78.4% higher gene loss rate than *O. rufipogon*, which may have reshaped the *O. sativa* into have a relatively stable genome. Whether this re-fitting process has contributed to its being selected as the domestication material is not clear and needs further exploration. Nonetheless, divergent gene loss rates among the *Oryza* plants over time suggest that the *Oryza* genomes may have undergone different levels of adaptive selection and/or genetic drift over different periods, likely to cope with changing environmental pressures.

### 2.4. Genome Instability Revealed by Characterizing Tandem Genes in Oryza Plants

DNA recombination is the key source of DNA mutation, which affects the stability of a genome. Ectopic DNA recombination may occur between tandem genes, inducing mutations among them and acting as a force to increase or decrease copy numbers of tandem genes. Here, we counted the tandem genes in *Oryza* plants and the referenced plant *L. perrieri*. The *Oryza* plants have 2647–5149 tandem genes, accounting for 8–16% of their respective total genes. *O. sativa* ssp. Jing has the most tandem genes (5248) among all *Oryza* plants, and the fewest was found in *O. barthii* (2647). Regarding the percentage of total genes, *O. punctata* has the most tandem genes among *Oryza* plants (Table 3).

Incurred genome instability due to tandem genes may be related to their distributive densities in a genome. Therefore, we characterized the average number of tandem genes regarding genomic DNA length (Mb). *O. brachyantha* has the highest average number of tandem genes (437 genes per Mb), followed by *O. sativa* ssp. Jing, with 375 genes per Mb. The other *Oryza* plants have an average of 219–289 genes per Mb.

We then counted the tandem genes on each chromosome of the *Oryza* plants. In all ten *Oryza* plants and three other species (and *L. perrieri*), chromosome 11 consistently had the highest proportion of tandem genes, ranging from 25.5% to 17.2%. For example, in *O. sativa* ssp. Jing rice, chromosome 11 accounted for 20.8% of the tandem genes, while the tandem genes on the remaining chromosomes ranged from 9.7% to 18.3%. A closer check of this finding found its relation to the high gene loss rates of chromosome 11 in the studied plants shown above and revealed the highest enrichment of NBS-LRR disease resistance genes, which are often a key set of tandem genes in these chromosomes, as further described below.

Disease resistance genes are extremely important for plant growth and the development process. *O. barthii* had the largest number of NBS-LRR disease resistance genes (644), while *O. brachyantha* had the fewest (409) (Table 4). Comparatively, the cultivated plant, *O. sativa* had a low number of NBS-LRR genes (492). In each plant, including the referenced one, NBS-LRR genes were significantly associated with tandem homologs (Table 4).

There are many NBS-LRR disease resistance genes in the tandem repeat genes of *Oryza* plants. For example, there are 135 NBS-LRR disease resistance genes in the tandem repeat genes of Jing rice. Compared with the 40,701 annotated genes in Jing rice, NBS-LRR resistance genes in tandem were overrepresented (chi-squared test *p*-value = 6.73 × 10^−22^), which may indicate that tandem duplication is an important driving force for the amplification of NBS-LRR disease resistance genes in rice. The number of NBS-LRR disease resistance genes in the tandem repeat genes of other *Oryza* plants is between 66 and 213. Except for *O. meridionalis*, the chi-squared test results are all less than 0.05. Based on the above data, this may indicate that tandem duplication is an important amplification driving force for NBS-LRR disease resistance genes in *Oryza* plants.

### 2.5. KEGG Enrichment Analysis of Genes

KEGG enrichment analysis was used to explore the functions of genes or proteins by mapping them to known biological pathways to reveal their potential roles [18]. This helped elucidate the functional relationships of genes and the molecular mechanisms underlying biological processes. We performed KEGG enrichment analysis for homologous genes lost in *O. sativa* (Jing rice) and *O. rufipogon* (Figure 4c,f). Most of the lost genes in *O. sativa* (Jing rice) were related to genetic information processing, with 627 genes accounting for 37.52% of the total. Similarly, in *O. rufipogon*, 610 lost genes were related to genetic information processing, accounting for 40.08% of the total. In Jing rice, 134 lost genes (8.02%) were associated with environmental information processing, while in *O. rufipogon*, 104 lost genes (6.83%) were classified into this category.

Regarding the genes lost after the split of *O. sativa* (Jing rice) and *O. rufipogon* and the genes specifically lost in each genome, in *O. sativa* (Jing rice), the lost ones were mostly related to genetic information processing and lipid metabolism (38.1% and 6.2% of the total specifically lost genes), respectively. In addition, the genes related to environmental information processing and signaling and cellular processes accounted for 9.9% and 8.3% of the total genes, respectively. A similar observation was made for *O. rufipogon* (Figure 4a,d). The genes related to genetic information processing and lipid metabolism in *O. sativa* (Jing rice) were 5.3% and 4.5% higher than those of *O. rufipogon*, respectively. In comparison, genes related to environmental information processing and signaling and cellular processes were 2.5% and 1.9% lower than those of wild *Oryza*, respectively.

Chromosome 11 of *O. sativa* (Jing rice) has a higher rate of gene loss than any other chromosome and the highest number of tandem repeats [19]. KEGG enrichment analysis showed that 28.6% of the tandem repeat genes on chromosome 11 of *O. sativa* (Jing rice) were related to carbohydrate metabolism and a similar percentage 27.08% of the carbohydrate metabolism were lost from the chromosome. In contrast, the genes lost from chromosome 11 of *O. sativa* (Jing rice) were less related to genetic information processing (4.1%) and more related to the biosynthesis of secondary metabolites (14.3%) (Figure 4b,e).

The above findings may reflect an adaptive strategy in the evolution of *Oryza* plants. This strategy may involve reducing redundant genes related to their growth and development, increasing genes conducive to their survival advantages in nature, and enhancing their resistance to various pressures in the growth environment.

### 2.6. Analyses of Other Poaceae Plants

Because interesting findings were obtained in *Oryza* species, we extended our research to other Poaceae plants, including *Pennisetum americarum*, *Setaria italica*, and *Sorghum bicolor*, which were affected by the GCT but not by extra WGDs. Within the GCT-produced homoeologous blocks, they had average numbers of collinear genes compared to the *Oryza* plants, showing comparable genome stability (Appendix A). Additionally, a cross-subfamily comparison to *L. perrieri* showed relatively fewer collinear genes in orthologous blocks, but S. bicolor and S. italica had higher average numbers of collinear genes than those in *O. rufipogon*, possibly due to the incompleteness of the genome sequence of the latter (Appendix A).

*P. americarum* had the fewest tandem genes per million base pairs, while *S. italica* and *S. bicolor* had 3–5 times more than those in *O. sativa* ssp. Jing rice, and even more than those in Xian rice. This shows that the genomes of *S. italica* and *S. bicolor* were not stable as indicated by the burst and expansion of tandem duplicated genes (Appendix A).

## 3. Discussion

The present study provides new insights into the evolution of *Oryza* and other Poaceae genomes, particularly with regard to the gene retention/loss of homologous genes produced by the GCT or the genes revealed to be orthologous to the *L. perrieri*, the accumulated or lost tandem genes in each genome. The findings display genome instability in different aspects.

Both the intragenomic and intergenomic analyses above show the evolutionary instability of *Oryza* genomes and their plasticity, resulting in rather varied gene (and DNA) contents. The GCT-produced homologous regions in the studied *Oryza* plants show divergent levels of genome fractionation, showing extensive DNA reshuffling after their split. The nearly 30% difference in homologous block numbers and the 35.7% difference in the collinear gene numbers per block indicate that certain *Oryza* genomes have been subjected to much more genome DNA permutations than others. These intragenomic findings were further enhanced by the comparative analysis of *Oryza* plants to the referenced plant, *L. perrieri*. The referenced plant shared the GCT with the *Oryza* species, making it a good reference to find the genomic changes after the ancestral tetraploidization. Regarding the intragenomic and intergenomic analyses of the numbers of collinear genes and the homologous (paralogous or orthologous) blocks, *O. nivara* and *O. meridionalis* were among the plants with the fewest collinear genes and the most blocks, indicating that they are likely to have instable genomes. However, this inference should be addressed carefully, as different genome sequences might have been assembled to varying credible levels. It should be noted that *O. sativa* (Jing rice and Xian rice), the best-assembled genome due to its agricultural value, was often among the top but not always inferred as the most stable genome.

Different chromosomes have shown divergent evolutionary instability, while the orthologous chromosomes have shown similar features. Notably, the above findings suggest that chromosomes with smaller chromosome ordinal numbers often preserve more genes, while those with bigger ordinal numbers have undergone more gene losses. Around one-third of genes were lost from *Oryza* Chro 1–3, while more than half were lost from Chro 11 and 12. A consistent finding spanning different *Oryza* plants cannot be explained by particularity in genome formation. We should note that some of these *Oryza* plants diverged tens of million years ago, and others only tens of thousands of years ago. The fact that different chromosomes have rather different gene loss rates and the orthologous chromosomes have similar gene loss rates indicates that considerable fluctuation in gene loss among chromosomes has occurred during the independent evolution of different species.

Nonetheless, the seemingly unusual evolutionary phenomena must have an evolutionary cause. To our knowledge, *Oryza* Chro 11 and 12 were proposed to be two homoeologous (or paralogous) chromosomes produced by the GCT (Figure 5). Illegitimate recombination was inferred to have occurred after the GCT and has lasted for ~100 million years, even being ongoing between one of their terminal regions [20,21]. Evidence was found to support that illegitimate recombination between them has resulted in extensive gene losses (and elevated mutations) and explains, at least partially, the present finding of the extensive gene losses in Chro 11 and 12. Illegitimate recombination may have occurred between all homoeologous chromosomes in the early days after the GCT. When the homoeology pattern is reduced due to DNA inversion or re-shuffled chromosome formation due to illegitimate recombination or the accumulation of transposable elements, illegitimate recombination could be significantly inhibited [20], contributing to a stable genome that mainly executes the diploid heredity. Higher gene loss rates in Chro 8 and 9 could be similarly explained by their shared homoeology across almost nearly entire length [21], although Chro 8 acquired a short extra DNA segment at the termini of its short arm. The higher gene loss rates in Chro 11 and 12 could be additionally explained by the enriched presence of NBS-LRR genes, which make up around one-fourth and one-sixth of the total genes in each genome respectively. The NBS-LRR genes often form tandem clusters in these chromosomes and are expected to have undergone a birth-and-death or duplication-and-loss process due to ectopic recombination. Copy numbers could have reduced during evolution, although an increased number of genes could have also occurred, which is determined by the interactions between the plants and the environment. The higher gene loss rate in Chro 10 might be explained by illegitimate recombination with its homoeologous chromosomes or chromosomal segments. However, supportive evidence has to be found in further research.

Lower gene loss rates in chromosomes with small ordinal chromosome numbers could be explained by another aspect, that is, illegitimate recombination might have been inhibited in these in the early days after the GCT, resulting in their stable nature in gene composition. *Oryza* Chro 2 and 3 are each produced by the nested chromosome fusion of two ancestral chromosomes (Figure 5) [22], which inhibited illegitimate recombination between affected homoeologous chromosomes and chromosomal regions. The fusion of ancestral chromosomes formed bigger chromosomes, making it easier to identify them by microscope and name them as the first chromosomes. A lower gene loss rate in Chro 1 could be explained by a DNA inversion that affected nearly the entire short arm of its homoeologous chromosome, Chro 5, restricting possible illegitimate recombination between them [22].

Supporting findings were found in the other Poaceae plants observed in this study. In *S. color, S. italica*, and *P. anericarum*, genome repatterning has resulted reorganization of the chromosomes, causing the gene loss rates to not follow the patterns shown in the *Oryza* plants (Figure 5). However, when we identified their orthologous regions corresponding to *Oryza* plants, we found the expected gene loss patterns consistent with those described in Oryza chromosomes. For example, we found that Sb-Chro 5, the counterpart of an *Oryza* chromosome, lost the most of its genes compared to the other sorghum chromosomes (Appendix A).

Tandem genes are a key source of genome instability due to recombination between homologous copies at ectopic locations [23]. Recombination, normal or illegitimate, is the reason for DNA mutations [24]. Ectopic DNA recombination can produce duplicated copies of genes, through the shifted pairing of neighboring homologous regions [25]. This process is proposed to often be accompanied by DNA cross-over or gene conversion. Meanwhile, ectopic DNA recombination can also lead to the deletion of tandem genes, possibly through the activities of tailoring enzymes that remove unpaired DNA sequences. Thus, the existence, expansion, and reduction of tandem gene copies would contribute to genome (in)stability and plasticity. Regarding biological function, tandem genes, by providing redundant gene copies, may reduce the risk of functional loss mutations or enlarge the mutational gene pool to fight frequent environmental changes [26]. As noted above, many disease-resistance genes, like NBS-LRR genes, are tandem genes. Evolutionarily, like other duplicated genes, tandem genes may contribute to species’ adaptivity by enriching combinational changes in gene expansion and functional divergence, helping them respond to biotic or abiotic pressures and promoting species diversity. Here, we revealed around a twofold difference in tandem genes and their genomic densities between *Oryza* plants, showing divergent genome stability and plasticity within the *Oryza* genomes. Moreover, the numbers of NBS-LRR genes are significantly different among the *Oryza* genomes, displaying varying capabilities and mechanisms to combat potential pathogens from their respective living niches.

As discussed above, we characterized intra- and intergenomic collinear genes and tandem genes to find whether there are differences in genome stability and plasticity among *Oryza* plants. The intra- and intergenomic gene collinearity explorations resulted in rather similar observations. This can be expected, given that they were related to the measurement of genome fractionation from different viewpoints. In contrast, the characterization of tandem genes resulted in different observations. For example, *O. sativa* showed the highest enrichment of tandem genes, indicating instability due to having many tandem genes. In contrast, it was listed as the third or fourth in having the fewest collinear gene blocks, which shows that it has a relatively stable genome. Overall, genome stability and plasticity should be measured from different aspects. Previously, it was reported that to help maintain genetic diversity, plant genomes without recent polyploidization, and therefore lacking recent collinear genes, could be complemented by a burst of tandem genes.

The present efforts provide insights into the genome (in)stability of *Oryza* and other Poaceae genomes, which may contribute to the understanding of *Oryza* genome formation, evolution, and functional innovation, and aid future research aimed at improving rice cultivars for sustainable agriculture.

## 4. Materials and Methods

### 4.1. Plant Genome Data Materials

The genomic data of ten *Oryza* species and four other species, including *Oryza sativa* ssp. Jing (indica) [27], *Oryza glaberrima* [28], *Oryza brachyantha* [29], *Oryza rufipogon* [30], *Oryza nivara* [30], *Oryza glumaepatula* [31], *Oryza meridionalis* [31], *Oryza barthii* [31], *Oryza punctata* [30], *Setaria italica* [32], *Sorghum bicolor* [33], and *Leersia perrieri* [30], were downloaded from the Gramene database (https://www.gramene.org (accessed on 2 October 2023)). *Oryza sativa* ssp. Xian (Japonica) (Minghui 63, a Xian rice) [27] was downloaded from the NCBI (National Center for Biotechnology Information). *Pennisetum americarum* [34] was downloaded from the Icrisat database (https://cegresources.icrisat.org/ (accessed on 2 October 2023)) (Appendix A).

### 4.2. Multiple Sequence Alignment

BLASTP was used to find putative homologous genes by comparing protein sequences between any two genomes or within each genome. The E-value was set to 1 × 10^−5^, and relatively loose criteria were used to accommodate ancient duplicated genes produced by the GCT.

### 4.3. Calculation of Synonymous Nucleotide Substitutions

The Nei–Gojobori approach [35] implemented in WGDI [36] was used to estimate the nucleotide synonymous substitution rates (Ks) between hom(e)ologous genes. The number of divergences was estimated by Ks. Python3 scripts in WGDI were used to draw dot plots of the homologous gene.

### 4.4. Genome Collinearity and Visualization

We used the WGDI-icl module to perform collinearity analysis. The collinear regions within each genome or between genomes were characterized based on scores, statistical significance, and the number of homologous genes in collinearity [37]. Within each genome, Ks values (~0.6) were used to find duplicated genes produced by the GCT. Ks values were used to differentiate the GCT-produced paralogous genes and the orthologous genes. This classification is not difficult because the Ks between orthologous genes is often <0.3, as compared to ~0.6 between paralogs.

Using the *L. perrieri* genome as a reference, we built genome alignments with collinear genes among all genomes. In an Excel table of multiple columns, the first column was filled with all genes from the reference genome. Then, in the second column, a cell was filled with the ID of a gene if it was detected to be collinear to the corresponding cell in the first column. The cell was filled with a dot if a collinear gene was absent. We assigned the corresponding number of the other columns according to the inferred gene collinearity between the genomes. That is, the third column of the table was filled with the orthologous genes by checking the gene collinearity between the considered genome and that of the reference. Then, the fourth column was filled with orthologous genes from another *Oryza* plant. When each of the studied plants occupied a column, the GCT paralogs in the *L. perrieri* were used to fill another column to compare the collinearity to those (also the *L. perrieri* genes) in the first column. Then, the following columns were filled with the paralogous genes from each *Oryza* plant. Eventually, genome alignment of all *Oryza* plants was constructed and thereafter visualized using a Circos plot, which was created using a home-made Python3 program [38].

### 4.5. Calculation of Gene Retention Rate Through Homologous Genes

Using the -r module of WGDI and *L. perrieri* as the reference genome, we calculated the number of genes without collinear orthologs in 9 *Oryza* plant chromosome regions in the reference genome. In this way, the gene loss rates of the orthologous genes were obtained.

### 4.6. Phylogeny and Divergence Time Analysis

We used the gene functional analysis tool OrthoFinder v2.2.5 to construct the species phylogenetic tree [39]. Amino acid sequences from *Oryza* species and *L. perrieri* were included. OrthoFinder identified the homologous gene groups using the diamond comparison method [40]. Phylogenetic trees were constructed using FastTree 2.1. MEGA-X (version 12.0.12) [41] was used for alignment, tree editing, and beautification. The default parameters were used unless otherwise specified.

### 4.7. KEGG Enrichment

KEGG (Kyoto Encyclopedia of Genes and Genomes) [42] enrichment analysis was performed on the protein sequences, lost homologous genes, and tandem genes of *L. perrieri* and *Oryza* species to investigate the impact of tandem duplication and gene loss on genome plasticity and stability. The default parameters were used unless otherwise specified.

### 4.8. Identification of NBS-LRR Disease Resistance Genes

We performed HmmSearch3.0 [43] analysis to find NB-ARC (PF00931) protein-coding genes, with an E-value of 1 × 10^−10^, to obtain genes containing the NB-ARC domain. Then, we performed a BLASTP search using NBS-LRR [44] disease resistance genes from *Arabidopsis thaliana* against the whole genomes of the *Oryza* species and four other species, with an E-value of 1 × 10^−10^, to obtain homologous disease resistance genes containing the NB-ARC domain in *Oryza* that are homologous to those in *A. thaliana*. Finally, we used the genes present in both species as the inferred NBS-LRR disease resistance genes for the whole genomes of *L. perrieri*, *S. italica*, *S. bicolor,* and the ten *Oryza* species. The same approach was used to obtain the members of NBS-LRR disease resistance genes in tandem repeat genes.

## 5. Conclusions

Our analyses revealed dynamic genome instability in *Oryza* species shaped by differential gene losses and recombination following the grass-common tetraploidization (~100 Mya). Smaller-numbered chromosomes retained more ancestral genes due to reduced recombination, while larger-numbered chromosomes (e.g., 11–12) experienced elevated gene loss linked to prolonged homoeologous recombination. Divergent tandem gene densities further highlighted the varying instability of chromosomes. These findings, consistent across *Poaceae*, show how historical polyploidization and chromosomal architecture drive genome evolution, offering insights for crop improvement and stress adaptation studies.

## Figures and Tables

**Figure 1 ijms-26-04778-f001:**
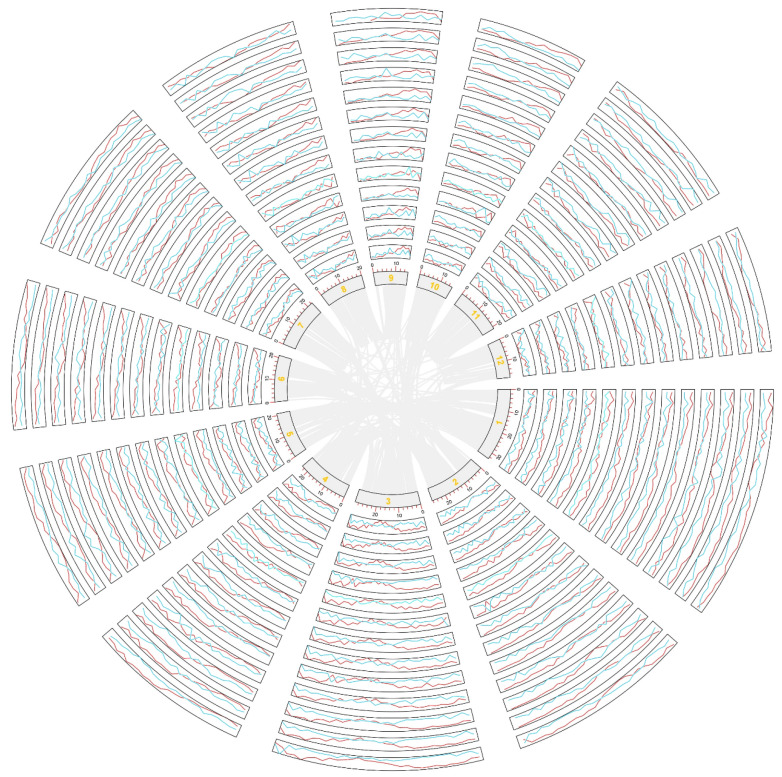
Loss and retention of genes on chromosomes among the studied species. *L. perrieri* was used as the reference genome, and a 1 MB sliding window was used to infer the collinear genes (red) and lost genes (light blue) in each chromosome of each species. Red represents retention, and blue represents loss. From the center, the chromosomes in the circle are from *O. barthii*, *O. brachyantha*, *O. sativa* Jing rice, *O. glaberrima*, *O. meridionalis*, *O. nivara*, *O. punctata*, *O. rufipogon, O. glumaepatula*, *O. sativa* Xian rice, *S. bicolor*, *S. italica*, and *P. americarum*. The orange numbers represent chromosome numbers.

**Figure 2 ijms-26-04778-f002:**
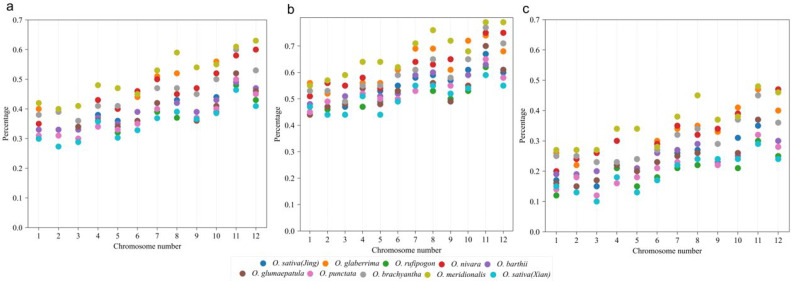
Gene loss rates in *Oryza* plants, with *L. perrieri* as the reference. The horizontal coordinate represents the 12 chromosomes of the *Oryza* plant, and the vertical coordinate represents the loss rates. The loss rates of 12 chromosomes of each species in the figure are represented by different colored dots. Note: (**a**) The loss rates of 12 chromosomes of each species. (**b**) The loss rates of 12 chromosomes of subgenome 1. (**c**) The loss rates of 12 chromosomes of subgenome 2.

**Figure 3 ijms-26-04778-f003:**
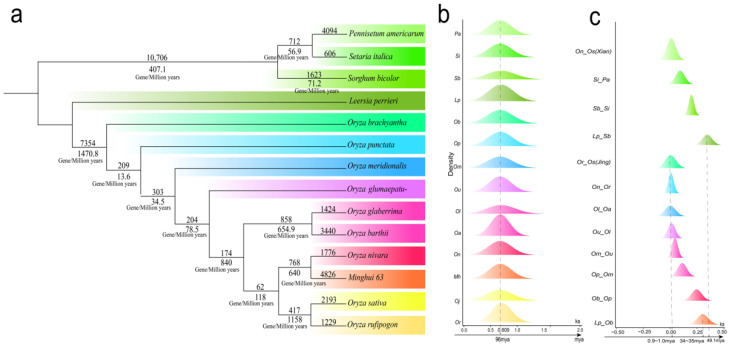
Phylogenetic relationship and gene loss rates during evolutionary periods. (**a**) Gene loss rate during the differentiation of *Oryza* plants. A segment of the branch has two numbers, with the top one showing the number of genes lost during the corresponding evolutionary period, and the bottom one showing the gene loss rate during the period. (**b**) The distribution of Ks values between collinear genes after evolutionary rate correction. (**c**) The Ks value of the divergence among species and the estimated divergence time. The abbreviations of the species are represented by the letters in the ordinate. Oa, Ob, Os (Jing), Ol, Om, On, Op, Or, Ou, Os (Xian), Sb, Si, and Pg stand for *O. barthii*, *O. brachyantha*, *O. sativa* Jing rice, *O. glaberrima*, *O. meridionalis*, *O. nivara*, *O. punctata*, *O. rufipogon*, *O. glumaepatula*, *O. sativa* Xian rice, *S. bicolor*, *S. italica*, and *P. americarum*.

**Figure 4 ijms-26-04778-f004:**
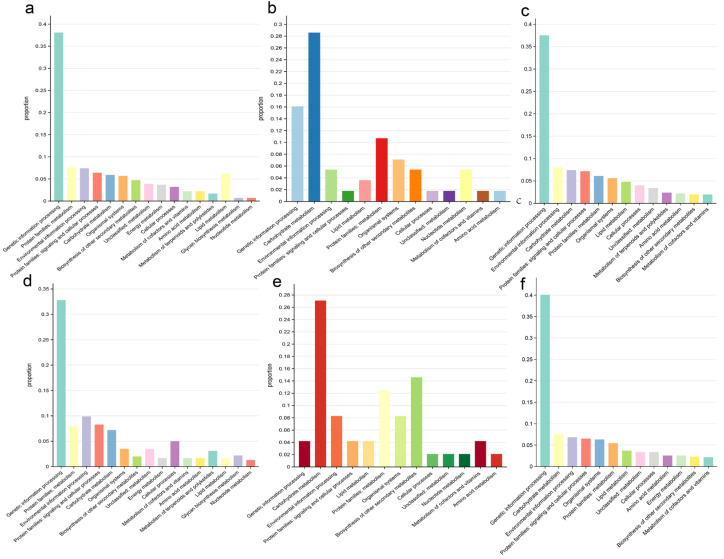
KEGG features of lost genes. (**a**) KEGG features of *Oryza sativa* Jing rice relative to *Oryza rufipogon*-specific orthologous genes. (**b**) KEGG features of tandem repeat genes on chromosome 11 of *Oryza sativa*. (**c**) KEGG features of homologous genes lost in *Oryza sativa* Jing rice. (**d**) KEGG features of *Oryza rufipogon* relative to Jing rice-specific orthologous genes. (**e**) KEGG features of lost orthologous genes on chromosome 11 of Jing rice. (**f**) KEGG features of homologous genes lost in *Oryza rufipogon*.

**Figure 5 ijms-26-04778-f005:**
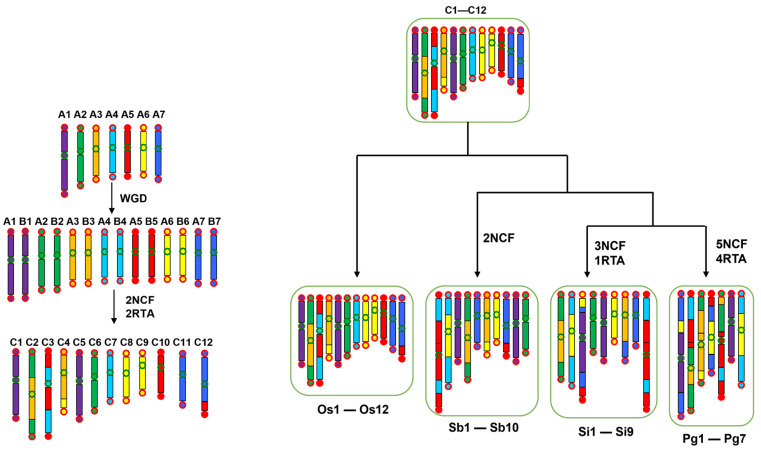
Evolutionary trajectory of grass chromosomes. The process of evolution from the ancestral chromosomes of grasses to those of rice, sorghum, foxtail millet, and pearl millet was reconstructed. Before the GCT, seven ancestral chromosomes (in different colors) were inferred. The duplicated chromosomes during the GCT were then reduced to 12 due to chromosome fusions and DNA exchanges, whose major structures have been well preserved in *Oryza* plants. The 12 chromosomes were then reduced in each plant under study. Os, Sb, Si, and Pg stand for *O. sativa*, *S. bicolor*, *S. italica*, and *P. americarum*.

**Table 1 ijms-26-04778-t001:** Homologous gene blocks and homologous genes within each *Oryza* plant and *L. perrieri*.

Species	Gene Number	Block Number	Gene/Block
*O. sativa* (Jing rice)	7914	348	22.74
*O. sativa* (Xian rice)	5922	292	20.28
*O. glaberrima*	7855	410	19.16
*O. rufipogon*	8387	348	24.10
*O. nivara*	7005	372	18.83
*O. barthii*	8254	323	25.55
*O. glumaepatula*	8081	345	23.42
*O. meridionalis*	6843	342	20.01
*O. punctata*	8848	370	23.91
*O. brachyantha*	7038	317	22.20
*L. perrieri*	7851	351	22.37

Note: Gene/Block represents the average number of collinear genes per homologous block inferred in each genome.

**Table 2 ijms-26-04778-t002:** Gene loss in orthologous regions in each *Oryza* plant relative to *L. perrieri*.

Species	Gene Number	Block	Gene/Block
*O. sativa* (Jing rice)	17,928	645	27.80
*O. sativa* (Xian rice)	19,203	1010	19.01
*O. glaberrima*	16,020	608	26.35
*O. rufipogon*	18,892	813	23.24
*O. nivara*	16,143	1295	12.47
*O. barthii*	18,036	729	24.74
*O. glumaepatula*	18,382	888	20.70
*O. meridionalis*	14,924	1085	13.75
*O. punctata*	18,722	798	23.46
*O. brachyantha*	16,407	499	32.88

Note: Gene/Block represents the average number of genes per orthologous block in each Oryza genome relative to *L. perrieri*.

**Table 3 ijms-26-04778-t003:** Tandem genes in *Oryza* plants.

Species	Whole Genome Length	Number of Tandem Gene	Tandem/Million Base Pair
*O. sativa* (Jing rice)	13.98	5248	375
*O. Sativa* (Xian rice)	25.66	7216	281
*L. perrieri*	17.34	4149	239
*O. barthii*	12.07	2647	219
*O. brachyantha*	11.30	4942	437
*O. glaberrima*	20.29	5051	248
*O. meridionalis*	18.36	4079	222
*O. nivara*	16.81	4541	270
*O. punctata*	17.79	5149	289
*O. rufipogon*	19.59	4409	225
*O. glumaepatula*	19.31	4703	243

**Table 4 ijms-26-04778-t004:** Numbers of NBS resistance genes in *Oryza* genomes.

Species	Gene	Tandem	NBS	N and t	*p*-Value
*L. perrieri*	29,114	4149	508	162	3.76 × 10^−30^
*O. barthii*	31,635	2647	644	91	1.41 × 10^−7^
*O. brachyantha*	34,588	4942	409	213	2.71 × 10^−106^
*O. sativa* (Jing rice)	40,701	5248	492	135	6.73 × 10^−22^
*O. glaberrima*	35,796	5051	422	91	1.34 × 10^−5^
*O. meridionalis*	29,442	4079	535	66	3.36 × 10^−1^
*O. nivara*	36,379	4541	637	164	3.07 × 10^−24^
*O. punctata*	31,791	5149	479	115	3.96 × 10^−6^
*O. rufipogon*	33,163	4409	643	186	8.83 × 10^−32^
*O. glumaepatula*	37,113	4703	595	188	4.34 × 10^−44^
*O. Sativa* (Xian rice)	57,129	7216	625	151	4.57 × 10^−18^

## Data Availability

If you need our dataset, you can contact the corresponding author.

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
