# Peer review of "Chromosome Ordinal Number-Related Genomic Stability Revealed Among Oryza and Other Poaceae Plants"

_ijms, 2025, doi:10.3390/ijms26104778_

Round 1
Reviewer 1 Report
Comments and Suggestions for Authors
Chromosome Ordinal Number-Related Genomic Stability Revealed Among Oryza and Other Poaceae Plants: the manuscript focused on the genomic stability of the studied genus considering other related genera and outgroups: the study seems to be with interesting scientific contribution. However, several points need to be updated/corrected:
Abstract:
1- L. perrieri : the authors should indicate the full scientific name of the plant species at least one time
2- The authors should also state clearly in the abstract the used method (briefly)
3- illegitimate recombination (line 21) did you think this is the suitable scientific terminology ?
4- A short sentence about the genus Oryza as an introduction in the abstract section : for example: Rice (Oryza sativa) is one of the key staple crops, providing food for nearly half of the
world's population......
5- Abbreviation in this section: NBS-LRR, Mya ...ect : the author should verify the policy of using acronyms and abbreviations (write the full text of each acronym at least on time in the beginning)
Introduction
Line: 43: 8 AA genome: abbreviations?
Line 55:
1-......genes have been lost in the rice genome after the GCT : add event after GCT
Material and methods:
2-Line 451: to 1e-5, and a relatively..... Please verify the correct way for 1e-5 writing
3-KEGG and many abbreviations regarding scientific data should be verified
4- Plant Genome Data Materials : could you please add a table summarizing: the plant species used material starting from genomic data bases
Results:
1- A short summary about the generated genomic data : Length, Alignment, %of GC..... should be presented
2- Table 1 : Gene/Block this is a ratio? the authors should state this clearly
3- Line 128 : To find gene retentioni : Please verify the correct writing of gene retention : You refer to gene retention index?
4- Line 132: Table 2: The numbers lost genes on the chromosomes of each Oryza plants using L. perrieri as a reference: Please rewrite the title of table
5- Line 136: ...... is much fewer than: please verify if the suitable ?
6- ............the most intable genomes are of O. nivara and O. meridionalis, while the stablest ones are of O. brachyantha........ On the basis of what did you conclud that?
7- Figure 1. : could you please ameliorate the quality of the figure
8. Figure 2: could you distinct any sub-groups on the basis of Gene loss rates ?
9- Figure 4 and 5: could you please ameliorate the resolution of the figure in order to make the figures more readable?
Discussion:
1- What about the chromosome number of the studied Oryza cultvars/species: the authors should mention that point in introduction and discuss the results with taking into account this detail
Conclusion :
Why this section is missed ?
Reviewer 2 Report
Comments and Suggestions for Authors
The manuscript entitled "Chromosome Ordinal Number-Related Genomic Stability 2
Revealed Among Oryza and Other Poaceae Plants" used genome of Leersia perrieri as the reference to compared genome stability in ten Oryza species and three other Poaceae species. The authors found that Oryza chromosomes with smaller ordinal numbers often preserved larger percentages of genes, while those with bigger numbers were subjected to more gene losses. This manuscript is well prepared and has enough data to support the conclusions. I would recommend accepting it for publication with few minor revisions.
1. in abstract, please use full name of L. perrieri.
2. Fig.1 is better to use one of the blank spaces to note each genome using the abbreviated name of each species that used in Fig. 3.
3. For the number of million years shown in Fig. 3 during evolution period of each species, citations are needed if they are from literature.
Round 2
Reviewer 1 Report
Comments and Suggestions for Authors
All comments are well answered and the paper in its current version can published
Author Response
Thank you for your valuable suggestions on our manuscript. We have revised the manuscript in accordance with your comments.